# A computational model to characterize the time-course of response to rapid antidepressant therapies

Abraham Nunes[1,2¤]*, Selena Singh[3]

**1** Department of Psychiatry, Dalhousie University, Halifax, Nova Scotia, Canada, **2** Faculty of Computer Science, Dalhousie University, Halifax, Nova Scotia, Canada, **3** Department of Psychology, Neuroscience & Behaviour, McMaster University, Hamilton, Ontario, Canada

¤Current address: 5909 Veterans' Memorial Lane, Abbie J. Lane Memorial Building (Room 3083B), QEII Health Sciences Centre, Halifax, Nova Scotia, Canada
* nunes@dal.ca

**Data Availability Statement:** All relevant data are within the paper and its Supporting information files.

## Abstract

Our objective is to propose a method capable of disentangling the magnitude, the speed, and the duration or decay rate of the time course of response to rapid antidepressant therapies. To this end, we introduce a computational model of the time course of response to a single treatment with a rapid antidepressant. Numerical simulation is used to evaluate whether model parameters can be accurately estimated from observed data. Finally, we compare our computational modelling-based approach with linear mixed effects modelling in terms of their ability to detect changes in the magnitude and time-course of response to rapid antidepressant therapies in simulated randomized trials. Simulation experiments show that the parameters of our computational model can be accurately recovered using nonlinear least squares. Parameter estimation accuracy is stable over noise levels reaching as high as 25% of the true antidepressant effect magnitude. Comparison of our approach to mixed effects modelling using simulated randomized controlled trial data demonstrates an inability of linear mixed models to disentangle effect magnitude and time course, while our computational model accurately separates these response components. Our modelling approach may accurately identify the (A) magnitude, (B) speed, and (C) durability or decay rate of response to rapid antidepressant therapies. Future studies should fit this model to data from real clinical trials, and use resulting parameter estimates to uncover predictors and causes of different elements of the temporal course of antidepressant response.

## Introduction

Depression is a major public health problem [1], for which there is growing interest in management using rapid antidepressant therapies [2]. These treatments, such as sleep deprivation [3], ketamine [4, 5], and classical psychedelics [6] are notable for their ability to produce robust antidepressant effects after only a single dose. The central problem with these therapies is currently the short duration of response [7].

**Funding:** Research Nova Scotia (AN) (grant: RNS-NHIG-2021-1931); QEII Foundation (AN). The funders had no role in study design, data collection and analysis, decision to publish, or preparation of the manuscript.

**Competing interests:** The authors have declared that no competing interests exist.

Understanding the time course of response to these novel antidepressant therapies is central to their development as evidence-based treatments for depression. We must study the predictors and causes of three aspects of treatment response: (1) the magnitude of response, (2) the speed of response (time to peak effect), and (3) the duration or persistence of response.

By predicting which patients are likely to respond significantly, quickly, and for a sustained period of time, we can better personalize treatment through optimized patient selection. By understanding the factors that predict or cause large, rapid, or sustained effects, we will be better able to develop strategies to potentiate or prolong rapid antidepressant action.

To discover factors that will predict the (A) magnitude, (B) speed, and (C) duration of response to rapid antidepressants, we must accurately measure these components of the overall time course, so that they can be used as dependent variables in predictive models. However, given the generally small sample sizes, resource intensity of experimental rapid antidepressant studies, and potential heterogeneity of response time courses across subjects, we must also balance the goal of characterizing temporal response profiles with the need to obtain sufficient statistical power. If we analyze rapid antidepressant response by comparing groups across multiple individual time points, we may not have sufficient statistical power due to multiple comparisons. Using computational modelling may improve statistical power in this setting [8].

Therefore, the present study introduces a nonlinear computational model to facilitate inference of the (A) magnitude, (B) peak response time, and (C) duration of response to rapid antidepressant therapies. We demonstrate that our model's parameters can be accurately estimated in practice, and outperform linear mixed-effects models in terms of capturing different aspects of the time course of rapid antidepressant response. This latter point is of significance due to the common use of linear mixed-effects models in the analysis of rapid antidepressant response data [5, 9–14]. Our statistical approach will support the development of real-world studies to uncover biomarkers for each phase of the antidepressant response time-course, so that recovery can be achieved faster, and sustained for longer periods of time.

## Materials and methods

Code to replicate all results of this study is included in S1 File.

### Computational model

We model the effect of a single administration of a rapid antidepressant at time $t$ as a normalized difference of exponentials, which is commonly used as a model of electrical conductance across postsynaptic membranes after arrival of discrete presynaptic action potentials [15]. Adopting this model for the present study is appropriate given its qualitative properties and relative simplicity. The model is formally defined as:

$$G_{g,a,b}(t) = \begin{cases} g\dfrac{\dfrac{e^{-\frac{t}{a}} - e^{-\frac{t}{b}}}{b}}{\left(\dfrac{b}{a}\right)^{\frac{b}{a-b}} - \left(\dfrac{b}{a}\right)^{\frac{a}{a-b}}} & \text{If } a \neq b \\[2em] \dfrac{gte^{1-\frac{t}{b}}}{b} & \text{Otherwise} \end{cases} \tag{1}$$

where $g$ is the magnitude of the peak effect (can be thought of as the magnitude of response) and $a > b$ are time constants. The constant $a$ is the "fall time," representing the amount of

time it takes for the magnitude of $G_{g,a,b}(t)$ to decline by a factor of e. We adapted the synaptic model in Eq 1 to model change in depressive symptoms, as measured by some rating scale, instead of measuring electrical current. This model was chosen because its three independent parameters ($g$, $a$, $b$) help disentangle the time course of response into its magnitude ($g$), speed ($b$), and duration ($a$).

The time until peak response can be computed as follows

$$t_{\text{peak}} = -\frac{ab \log\left(\frac{b}{a}\right)}{a - b}, \tag{2}$$

unless $a = b$, in which case $t_{\text{peak}} = a = b$. The effects of parameter alterations on the value of $G_{g,a,b}(t)$ and $t_{\text{peak}}$ are shown in Fig 1.

Using this model, one can show that a rapid *and* sustained response occurs as $a$ grows and $b$ approaches 0:

$$\lim_{b \to 0^+} \lim_{a \to \infty} G_{g,a,b}(t) = g, \tag{3}$$

whereby the response becomes a step function.

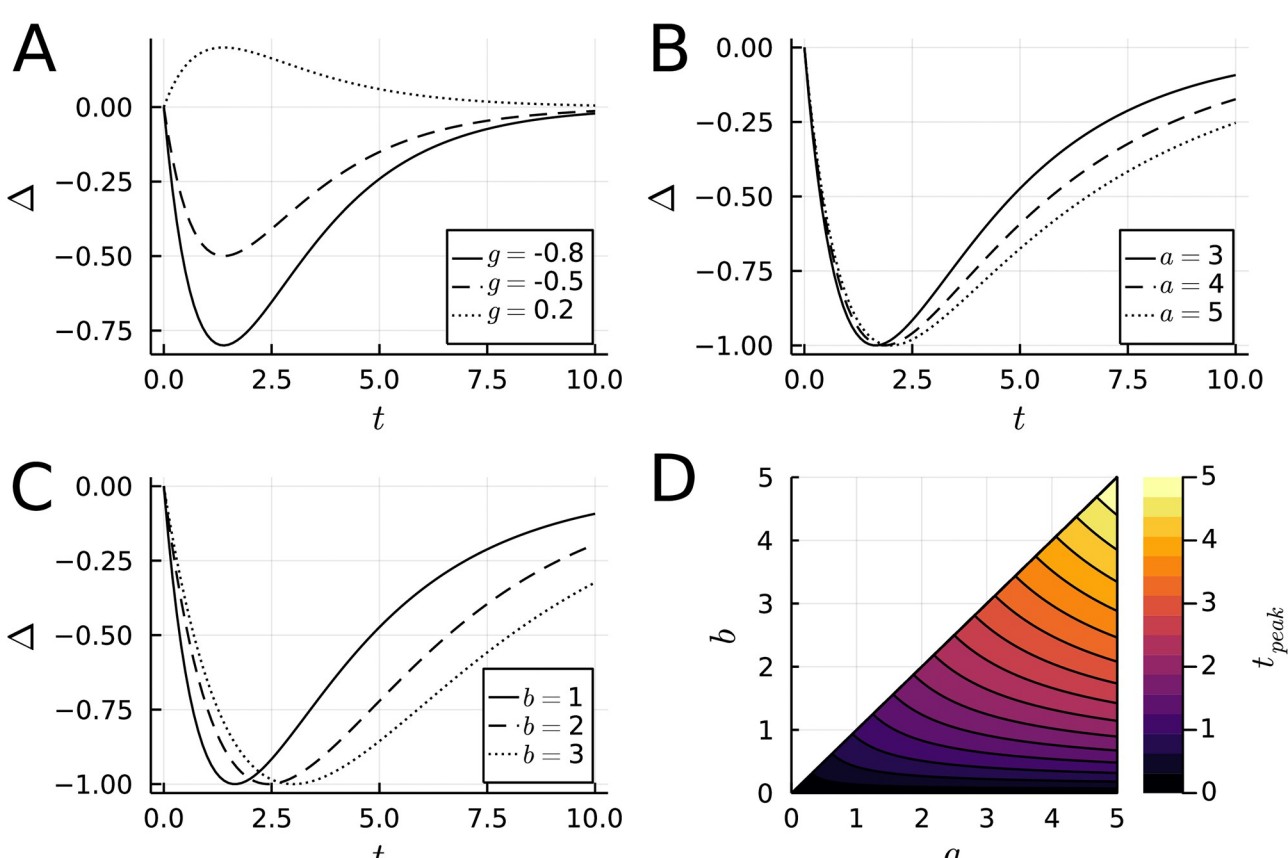

**Fig 1. Difference of exponentials model of ketamine response.** Time $t = 0$ in all cases represents the time of administration, and the units can be set arbitrarily by experimenters. $\Delta$ denotes the change in symptoms, conventionally measured using a rating scale. The default parameters for the model in each figure were set to $g = -1$, $a = 1$, $b = 0.5$.

## Simulation and parameter estimation

We first sought to determine whether, given simulated antidepressant response trajectories, the correct underlying parameters could be recovered statistically. To do this, we first simulated noisy antidepressant response trajectories as follows:

$$\tilde{G}_{g,a,b,\sigma}(t) = G_{g,a,b}(t) + \epsilon(t), \tag{4}$$

where $\epsilon(t) \sim \mathcal{N}(0, \sigma)$ is Gaussian noise with standard deviation $\sigma > 0$.

We fit the curve $G_{g,a,b}(t)$ (Eq 1) to a simulated observed symptom response curve $Y = (y_t)_{t=1,2,\ldots,T}$ using nonlinear least squares. Formally, this problem is defined as

$$(\hat{g}, \hat{a}, \hat{b}) = \operatorname{argmin}_{g,a,b} \sum_{t=1}^{T} [y_t - G_{g,a,b}(t)]^2, \tag{5}$$

and can be solved using widely available software packages for nonlinear optimization. In this study, we use the `Optim.jl` package for the Julia Programming Language (v. 1.8.2) [16]. We fit our model to samples of 100 simulated subjects whose response profiles were sampled according to the following parameters:

$$g \sim \text{Uniform}(-1, 1) \tag{6}$$

$$b \sim \text{Uniform}(0.01, 4) \tag{7}$$

$$\delta_a \sim \text{Uniform}(0, 5) \tag{8}$$

$$a = b + \delta_a \tag{9}$$

where Uniform($l$, $h$) denotes a uniform distribution with lower bound $l$ and upper bound $h$. We repeated this across 25 increasing levels of noise $\sigma \in \{0.001, 0.011, 0.021, \ldots, 0.241\}$. Note that the maximal noise level $\sigma_{max} = 0.241$ corresponds to an average variation of $\approx 25\%$ of the maximal effect $|g| = 1$. Parameter ranges were chosen to capture values that may be relevant to real world data. For instance, an effect magnitude of $-1 \leq g \leq 1$ captures the possibility of improvement ($g < 0$), no change ($g \approx 0$), and worsening ($g \geq 0$). The ranges of $a$ and $b$ were chosen to allow for a $t_{\text{peak}}$ between 0 and 6 (assuming the relevant time unit is days). We must also maintain the fundamental constraint of $a > b$.

We quantified the error in parameter estimation across these noise levels using the median absolute deviation (MAD) and interquartile range.

## Comparison to linear mixed model

We then sought to evaluate the statistical power of our modelling approach (Eqs 1 and 5) compared to that of linear mixed effects modelling, which is a commonly used analytical strategy in rapid antidepressant research capable of capturing repeated measures data within subjects, while identifying group-level variation [5, 9–14]. To do this, we simulated a large number of two-group parallel designed randomized trials with 20 subjects per group, in which one group demonstrated an average peak effect of $g^{(0)} = -0.5$, rise time constant of $b^{(0)} = 1$ day, and a decay time of $a^{(0)} = 5$ days. Comparison groups were constructed by systematically changing ($g$, $a$, $b$) by gradually larger amounts. We simulated daily ratings over a 21 day observation period, to model a time horizon and measurement frequency that relevant to the real-world use of rapid antidepressants [5, 9, 17, 18].

When varying one of the parameters ($g$, $a$, $b$), respectively, in the comparison group, all others were held equal to that in the control group. The following perturbation levels were tested: $\delta_b \in \{0.25, 0.5, \ldots, 3.0\}$, $\delta_g \in \{0.01, 0.03, \ldots, 0.19\}$, and $\delta_a \in \{0.25, 0.5, \ldots, 3.0\}$. The comparison group parameter values were subsequently computed as $b^{(1)} = b^{(0)} + \delta_b$, $g^{(1)} = g^{(0)} + \delta_g$, and $a^{(1)} = a^{(0)} + \delta_a$.

We implemented linear mixed effects modelling as a baseline comparator method. In R syntax, the mixed effects model was defined as

$$Y \sim Group*Time + (1|Subject). \tag{10}$$

After inferring the individual level parameters for each subject using nonlinear least squares (Eq 5), denoted $(\hat{g}, \hat{a}, \hat{b})$, we tested for group differences using the following mixed effects model

$$z \sim Group + (1|Subject). \tag{11}$$

for $z \in \{\hat{g}, \hat{a}, \hat{b}\}$. For each perturbation level, we ran 10 simulated experiments. Statistical power was computed for each regression coefficient as the proportion of runs where the p-value of the estimate was under $\alpha = 0.05$. Inference of all mixed-effects models was implemented using the `MixedModels.jl` (v4.8.0) package for the Julia Programming Language. We ran the entire experiment on an Apple M1 Max processor (2021; Apple Inc.; Cupertino, CA) in parallel over 16 threads.

## Results

### Parameter recovery experiment

Results of our parameter recovery experiment are shown in Figs 2 and 3. For low levels of noise, one can observe that the parameter recovery is highly accurate (Fig 2), and increases slowly (sublinearly) for increasing levels of noise (Fig 3). Specifically, at the maximal level of noise (corresponding to 25% of the peak effect magnitude), the median absolute error in estimation of $g$ was under 0.15. Similarly, for parameter estimates of $a$ and $b$, respectively, the median absolute error levels demonstrate a sub-linear increase with respect to noise levels, suggesting that parameter estimates are relatively stable to noise levels in the underlying data-generating process.

### Comparison to linear mixed Model

Using our modelling approach to disentangle variation in components of the time-course of rapid antidepressant response generally showed higher statistical power than the typical repeated-measures linear modelling approach (Fig 4).

Our modelling approach was better able to detect accurate group level differences across all levels of variation for all parameters compared to the classical modelling approach. For variation in $a$ and $b$, one should expect that the classical modelling approach should reveal a *Group × Time* interaction effect, but not a pure *Group* effect. However, the *Group × Time* was only statistically significant at extremely large perturbation values, and independent *Group* effects were detected early. This is a problem, because perturbation of $a$ and $b$ change the *time course* of response, but not the magnitude of response.

Fig 4 shows that our model-based approach detected group-level differences in variation of $a$, $b$, and $g$, with greater statistical power than the classical approach across virtually all perturbation magnitudes (see plots along leading diagonal of Fig 4, from top left to bottom right). Furthermore, there is very limited risk of false positive results, compared to the mixed

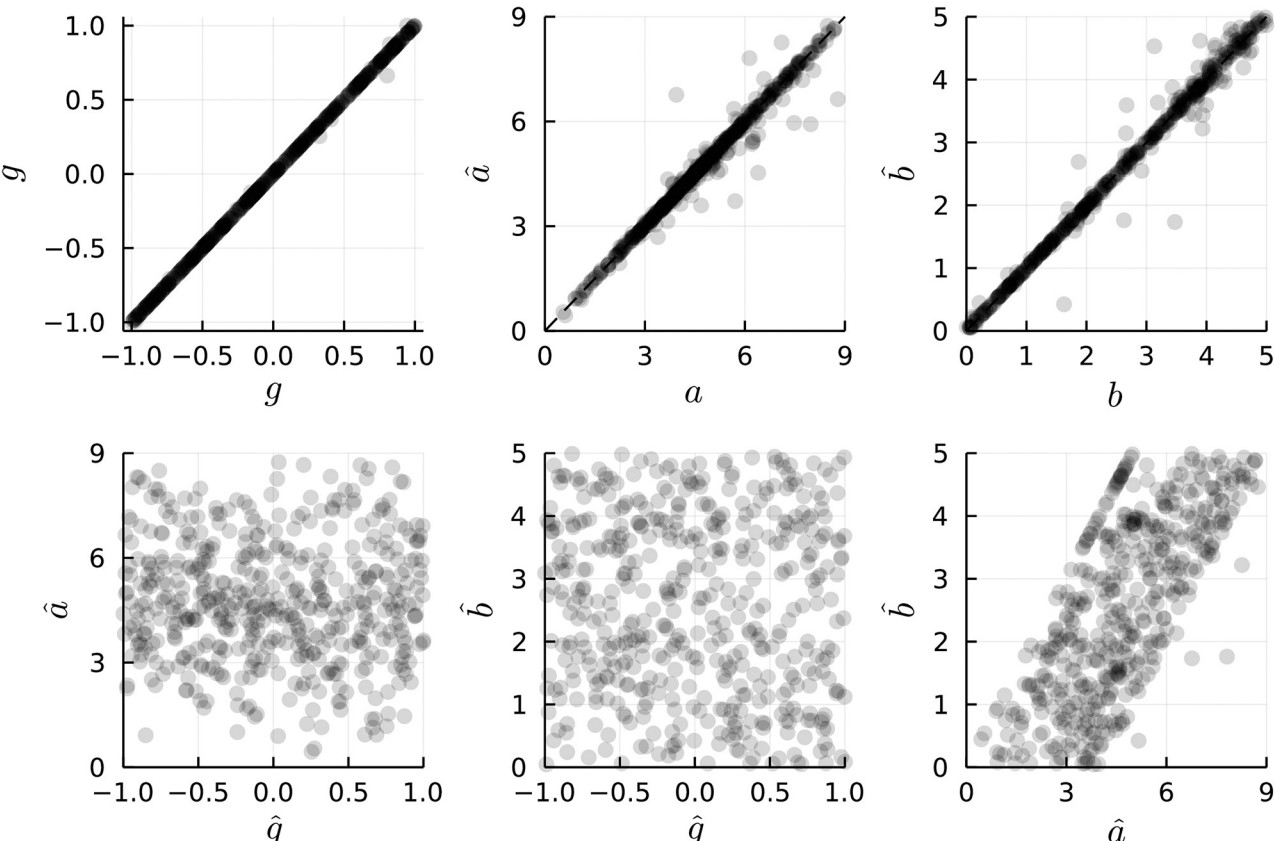

**Fig 2. Results of parameter estimation experiment.** The top row demonstrates ground truth (x-axis) vs. estimated (y-axis) parameters. The dashed line along the diagonals indicate the point of perfect parameter recovery. The lower row of plots depict relationships between parameter estimates, to verify absence of correlation in parameter estimates. Note that since the models are constrained such that $\hat{a} > \hat{b}$, the plot of $\hat{a}$ vs. $\hat{b}$ demonstrates what appears to be a positive correlation.

modelling method. For instance, group level differences in time course parameters ($a$, $b$) are rarely detected when the true underlying variation between groups occurs solely in the magnitude of effect (middle row in Fig 4). One also observes rare detection of statistically significant group differences in $g$ (the magnitude of effect) when the true underlying differences between groups are time course-related (i.e. in $a$ and $b$).

## Discussion

We have introduced a computational model that can be used to disentangle different components of the time course of rapid antidepressant response: the magnitude, speed, and stability or decay of the effect over time. This model is simple and its parameters are interpretable (magnitude of effect $g$, decay time $a$, response time $b$). Furthermore, these parameters can be accurately recovered from data, and these estimates are robust to noise. Finally, we have demonstrated that an existing standard approach using linear mixed models cannot separate overall antidepressant effects from changes in time course, whereas our model can accurately separate these components. Thus, our approach may be useful for future studies that aim to study predictors, causes, and modifiers of the temporal course of response to rapid antidepressant therapies.

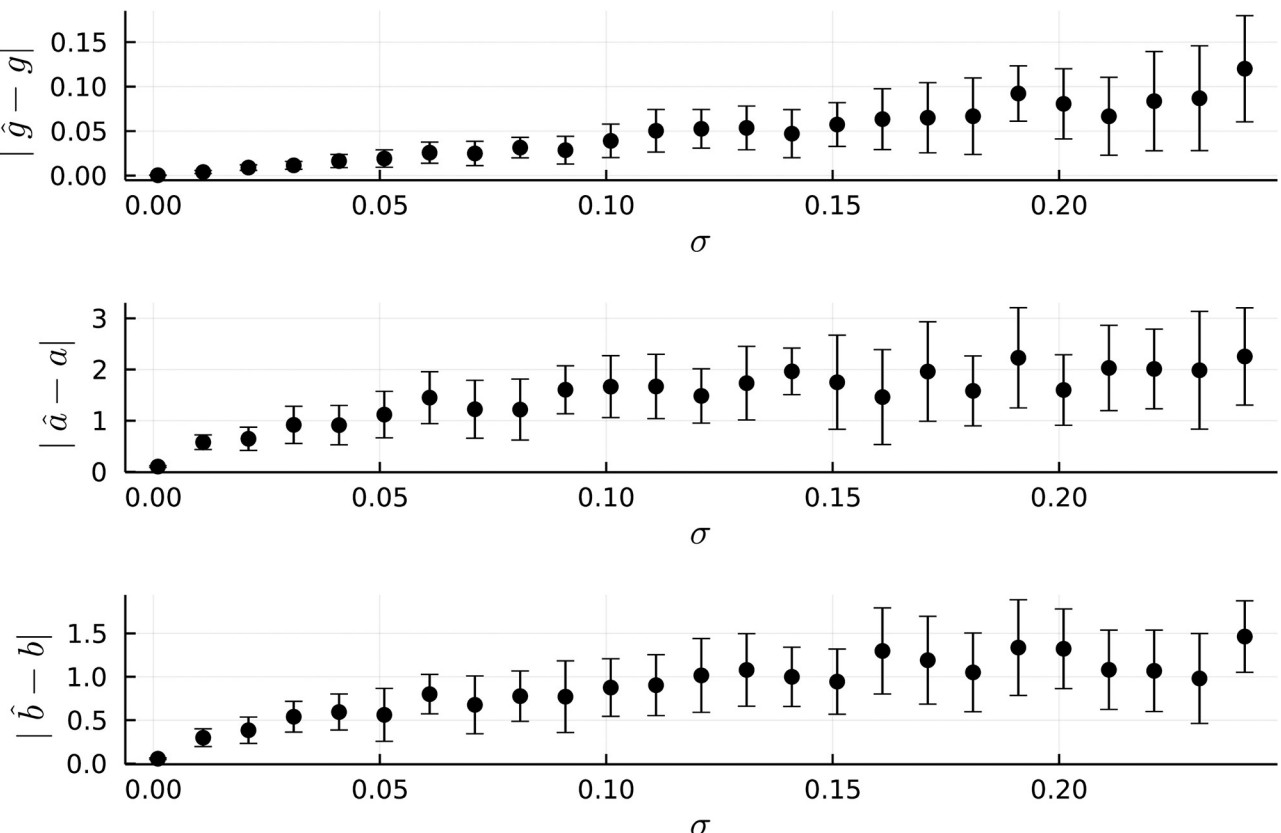

**Fig 3. Median absolute parameter estimation error in relation to response trajectory noise.** Error bars are interquartile ranges.

Two major questions about rapid antidepressant therapies concern (1) patient selection and response prediction [19], and (2) prolonging treatment effects [7]. Many investigations are being undertaken to identify predictors of ketamine's antidepressant effects, given the rapidly widespread use of this treatment modality [19, 20]. Several predictors of response to ketamine have been identified, including intra-infusion dissociation [21], fewer treatment failures [22], family history of alcohol dependence [23], anhedonic or interest-activity symptoms [24], obesity [25], and mixed/anxious features of depression [26]. Electroencephalographic markers have also been identified, most notably increases in gamma power [20]. There is much less evidence available concerning predictors of response to other rapid antidepressant therapies, such as sleep deprivation or psychedelics.

To our knowledge, none of the aforementioned predictors have been specifically associated with either magnitude, speed, or duration of response to ketamine. Yet, it is important to stratify predictors based on which features of the antidepressant response trajectory they predict. It is perhaps most important to find predictors of the duration/durability of rapid antidepressant response (which is encoded in our model using the parameter $a$). We must also seek to discover adjunctive or maintenance therapies for rapid antidepressants. Using our model-based approach, the effectiveness of these adjunctive or maintenance treatments can be measured directly based on their ability to prolong decay time $a$. Therefore, future studies should evaluate the degree to which biomarkers are specifically predictive of the magnitude ($g$), speed ($b$), and duration ($a$) of response to rapid antidepressant therapies. By doing so, we may better

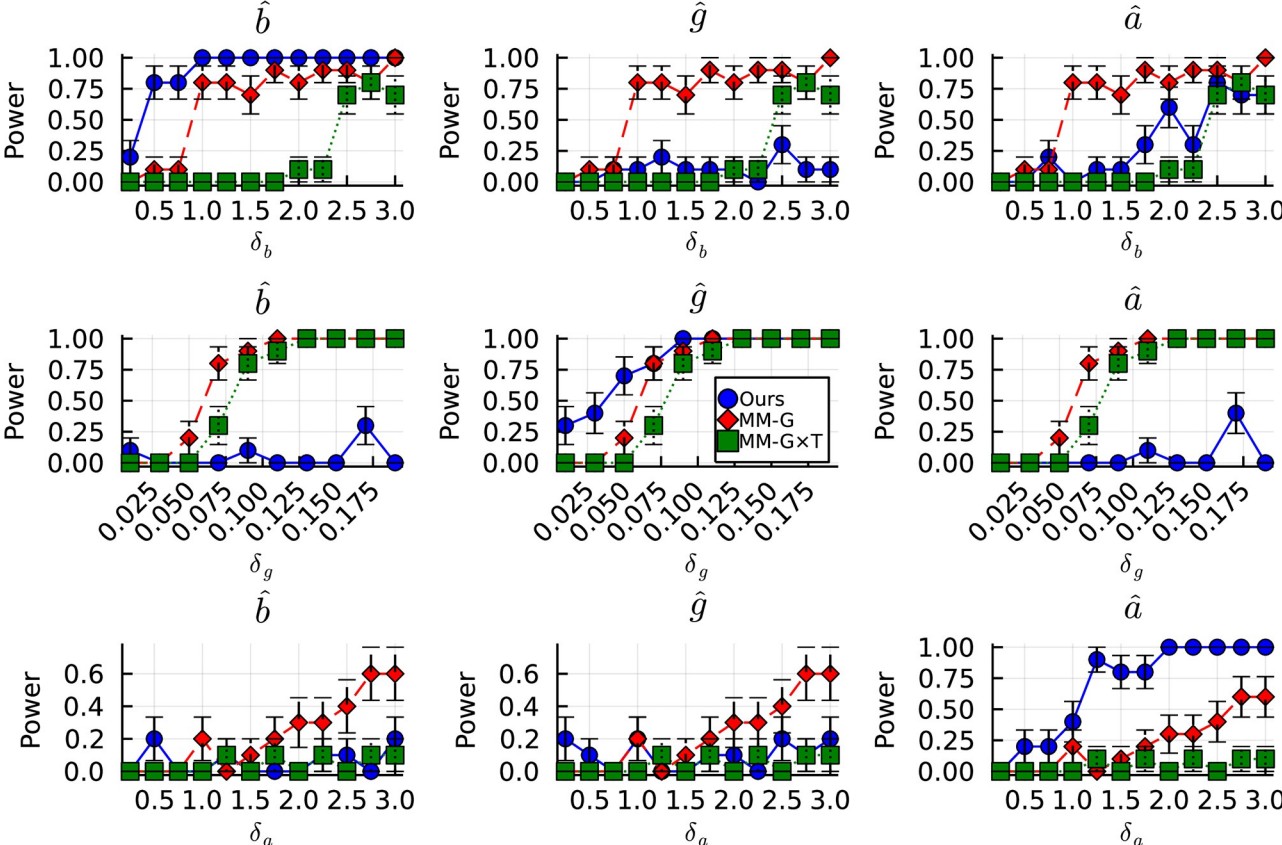

**Fig 4. Statistical power of our model compared to mixed effects modelling of the time course of rapid antidepressant effects.** X-axes denote the magnitude with which control-group parameters were perturbed to establish a comparator (e.g. "Intervention") group: $\delta_g$ is the difference in peak response magnitude, $\delta_a$ is the difference in decay time, and $\delta_b$ is the difference in rise time. Y-axes denote the proportion of runs in which statistical models detected statistically significant effects (Eqs 10 and 11). Each row depicts results for runs in which a specific parameter was varied (top row is for variation in $b$, middle row for variation in $g$, and the bottom row for variation in $a$). Each column (labeled in the title of each plot) indicates which parameter (using our method) is being tested for group level differences. **Blue** curves with circular markers represent the statistical power for group level differences in the parameters estimated using our approach (labeled "Ours"). **Red** curves with diamond markers represent the statistical power for *Group* effects detected using the classical mixed effects model (labeled "MM-G" in the legend). **Green** curves with square markers depict the statistical power for *Group × Time* effects detected using the classical mixed effects model approach (labeled "MM-G×T" in the legend).

personalize the application of rapid antidepressant therapies to the needs of individual patients. For example, patients may demonstrate clinical features that predict rapid response ($b$) to one antidepressant, but sustained response ($a$) to another antidepressant. Understanding these predictive markers may facilitate devising optimal personalized combinations for such patients, with one agent used for rapid termination of the depressive episode while the other is used for sustaining response.

One limitation of our paper is that all model-fitting experiments used maximum likelihood (point) estimation, which was employed for simplicity and conservatism. As such, the errors in our parameter estimation experiments are likely higher than if we had used maximum a posteriori or Bayesian estimation procedures constraining individual-level parameter estimates using the group-level mean and covariance.

Another limitation of our experiment is the lack of availability of real patient-derived data from rapid-antidepressant therapy trials. However, our group does not yet have access to such data. Furthermore, it is important to conduct internal validation of a modelling approach

under controlled circumstances, as we have done here, prior to application with real-world data.

Having demonstrated the ability of our model to disentangle different phases of a rapid antidepressant response trajectory, and having shown its specificity and sensitivity to noise, future studies can apply this model to real-world data. Therefore, future studies should aim to fit our difference of exponentials model to data derived from patients undergoing rapid antidepressant therapies in randomized trials. By regressing the parameters ($g$, $a$, $b$) against various predictors and biomarkers, such studies will be able to discover factors that influence the magnitude, speed, and duration of rapid antidepressant response.

## Conclusion

We have developed a computational model of the time course of response to rapid antidepressant therapies. This model can be fit to observed response time-series data in order to disentangle elements responsible for the magnitude, speed, and duration of response. Future studies should apply this model to real world data in order to uncover predictors and causes of these different temporal components of rapid antidepressant response.

## Supporting information

**S1 File. Code to reproduce the experiments and figures.**
(ZIP)

## Author Contributions

**Conceptualization:** Abraham Nunes.

**Data curation:** Abraham Nunes.

**Formal analysis:** Abraham Nunes, Selena Singh.

**Funding acquisition:** Abraham Nunes.

**Investigation:** Abraham Nunes.

**Methodology:** Abraham Nunes, Selena Singh.

**Project administration:** Abraham Nunes.

**Resources:** Abraham Nunes.

**Software:** Abraham Nunes.

**Validation:** Abraham Nunes, Selena Singh.

**Writing – original draft:** Abraham Nunes.

**Writing – review & editing:** Abraham Nunes, Selena Singh.

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
