## [Decision Letter · Decision Letter 0]

14 Dec 2023

PONE-D-23-38485A Computational Model to Characterize the Time-Course of Response to Rapid Antidepressant TherapiesPLOS ONE

Dear Dr. Nunes,

Thank you for submitting your manuscript to PLOS ONE. After careful consideration, we feel that it has merit but does not fully meet PLOS ONE’s publication criteria as it currently stands. Therefore, we invite you to submit a revised version of the manuscript that addresses the points raised during the review process.

We look forward to receiving your revised manuscript.

Kind regards,

Souparno Mitra, M.D.

Academic Editor

PLOS ONE

Journal Requirements:

   "Research Nova Scotia (AN); QEII Foundation (AN)"

Additional Editor Comments:

**Please see underlying reviewer comments and address these comments to consider for publication approval. **

Reviewers' comments:

Reviewer's Responses to Questions

**Comments to the Author**

1. Is the manuscript technically sound, and do the data support the conclusions?

Reviewer #1: Yes

Reviewer #2: Yes

2. Has the statistical analysis been performed appropriately and rigorously? 

Reviewer #1: Yes

Reviewer #2: Yes

3. Have the authors made all data underlying the findings in their manuscript fully available?

Reviewer #1: Yes

Reviewer #2: Yes

4. Is the manuscript presented in an intelligible fashion and written in standard English?

Reviewer #1: Yes

Reviewer #2: Yes

5. Review Comments to the Author

Reviewer #1: Overall, the research paper you summarized appears to have made a significant attempt at developing a computational model that can enhance the understanding of responses to rapid antidepressant therapies.

The innovation of the computational model is a strong aspect of the paper. The claim of superiority rests on the paper's ability to demonstrate tangible advantages through the simulation results. This argument would be stronger with a broader comparison to multiple established models in the field, perhaps through a meta-analysis or systematic review, to give context to the model's performance relative to a wider spectrum of existing methodologies. The integration of computational modeling within the clinical context presents compelling utility. Nevertheless, the strength of this argument would be magnified by offering a roadmap outlining how the model's outcomes will translate into personalized medicine. For example, the paper might showcase how specific response profiles identified by the model can inform tailored treatment decisions, such as dosage adjustments or the choice of therapeutic interventions. The paper seems to present a distinctly original approach through the design and implementation of this predictive model.

Reviewer #2: Dr. Souparno Mitra, MD

PLOS ONE

Dear Dr. Souparno Mitra,

I have had the honor of reviewing the article titled "A Computational Model to Characterize the Time-Course of Response to Rapid Antidepressant Therapies” submitted to PLOS ONE for peer review.

The manuscript addresses a very interesting topic, authors introducing a computational model that they claim can accurately identify magnitude, speed and durability or decay rate of response to rapid antidepressant therapies. The author's further claims that they have demonstrated that an existing standard approach using linear mixed models cannot separate overall antidepressant effects from changes in the time course whereas their proposed model can accurately separate these components which can also be considered as a major highlight of this article. The authors in this article claimed that their proposed modeling approach is able to disentangle the variations in the components of the time course of rapid antidepressant response and has showed greater statistical power than the typical repeated measures linear modeling approach, the authors added that their proposed modeling approach was able to better detect accurate group level differences across all levels of variation for all parameters compared to the classical modeling approach.

Some of the major highlights of this article includes the authors proposed model being able to aid in understanding the time course of response to the newer rapid antidepressant therapies which have a shorter duration of response to establish them as evidence based treatments for depression. The authors proposed a nonlinear computational model as they claim has the edge of improving statistical power while comparing groups at multiple time points which is a very crucial data that the existing linear mixed effects models lack.

The article presented multiple limitations which the authors identified including that all model fitting experiments used maximum likelihood estimation which invalidated the variability in real scenarios, also the authors identified the lack of availability of real patient derived data from antidepressant therapy trials which can be a significant data point adding to the lack of validity to the authors claimed results of their models ability.

This opens a new avenue to study the different phases of novel therapies with higher statistical power while establishing their effectiveness but the current model with the above mentioned limitations needs more testing with more clinical data and comparisons to be considered as a valid tool to apply the findings and be considered valid for clinical application.

6. PLOS authors have the option to publish the peer review history of their article (what does this mean?). If published, this will include your full peer review and any attached files.

Reviewer #1: **Yes: **Aditi Sharma

Reviewer #2: **Yes: **Surya Karlapati MD

---

## [Author Response · Author response to Decision Letter 0]

18 Dec 2023

Please see attached file with responses to reviewers.

---

## [Decision Letter · Decision Letter 1]

11 Jan 2024

A Computational Model to Characterize the Time-Course of Response to Rapid Antidepressant Therapies

PONE-D-23-38485R1

Dear Dr. Nunes,

We’re pleased to inform you that your manuscript has been judged scientifically suitable for publication and will be formally accepted for publication once it meets all outstanding technical requirements.

Kind regards,

Souparno Mitra, M.D.

Academic Editor

PLOS ONE

Additional Editor Comments (optional):

Reviewers' comments:

Reviewer's Responses to Questions

**Comments to the Author**

1. If the authors have adequately addressed your comments raised in a previous round of review and you feel that this manuscript is now acceptable for publication, you may indicate that here to bypass the “Comments to the Author” section, enter your conflict of interest statement in the “Confidential to Editor” section, and submit your "Accept" recommendation.

Reviewer #1: All comments have been addressed

Reviewer #2: All comments have been addressed

2. Is the manuscript technically sound, and do the data support the conclusions?

Reviewer #1: Yes

Reviewer #2: Yes

3. Has the statistical analysis been performed appropriately and rigorously? 

Reviewer #1: Yes

Reviewer #2: Yes

4. Have the authors made all data underlying the findings in their manuscript fully available?

Reviewer #1: Yes

Reviewer #2: Yes

5. Is the manuscript presented in an intelligible fashion and written in standard English?

Reviewer #1: Yes

Reviewer #2: Yes

6. Review Comments to the Author

Reviewer #1: I commend the author for their swift and thorough response to the raised concerns. The revisions made have significantly improved the overall quality of the work, showcasing a keen attention to detail and a commitment to addressing the feedback constructively.

Reviewer #2: Thank you addressing the comments for the manuscript titled " A Computational Model to Characterize the Time-Course of Response to Rapid Antidepressant Therapies" submitted to PLOS ONE for peer review.

All the comments were appropriately addressed.

7. PLOS authors have the option to publish the peer review history of their article (what does this mean?). If published, this will include your full peer review and any attached files.

Reviewer #1: **Yes: **Aditi Sharma

Reviewer #2: No

---

## [Editor Report · Acceptance letter]

25 Jan 2024

PONE-D-23-38485R1 

PLOS ONE

Dear Dr. Nunes, 

I'm pleased to inform you that your manuscript has been deemed suitable for publication in PLOS ONE. Congratulations! Your manuscript is now being handed over to our production team.

Kind regards, 

on behalf of

Dr. Souparno Mitra 

Academic Editor

PLOS ONE